# OpenReview forum: "Exposing Weak Links in Multi-Agent Systems under Adversarial Prompting"
_ICLR.cc/2026/Conference — Submitted to ICLR 2026_

### Official Review · Reviewer_v7hB · 2025-10-27

**Soundness:** 2
**Presentation:** 2
**Contribution:** 2
**Rating:** 2
**Confidence:** 3

**Summary:**

The paper introduces SAFEAGENTS, a unified framework for evaluating the safety of multi-agent systems (MAS) under adversarial prompting. It proposes a new diagnostic metric called DHARMA that classifies failures across planner and sub-agent levels, aiming to identify “weak links” in different architectural designs. The authors evaluate different representative MAS frameworks across four safety benchmarks.

**Strengths:**

The topic is timely and relevant as multi-agent systems are increasingly being deployed in real-world applications, yet their safety remains underexplored.
The experiment is also comprehensive, covering multiple architectures and benchmarks.

**Weaknesses:**

The writing requires significant improvement.

For instance, in lines 301–303, the authors write: “Planner-Failed identifies a critical failure mode where the planner does not refuse but fails to generate a valid executable plan, yet the system continues execution despite a valid plan.” This sentence is self-contradictory — the planner “fails to generate a valid plan,” yet the text later says “despite a valid plan.” Presumably, the authors meant “an invalid plan.” Such inconsistencies seriously affect readability.


Line 356: “for example” should not be capitalized mid-sentence.


Line 351: “ReAct” is mentioned without citation or definition; it appears abruptly without prior explanation or connection to Table 1.


Line 459: the abbreviation “SLM” is unclear — if it refers to Small Language Model, it should first be written in full when it appears for the first time. Furthermore, it is unclear why SLMs are introduced at all. Most experiments in the paper involve large, frontier models, so the inclusion of SLMs requires explicit motivation. If the authors believe that smaller models offer unique practical advantages (e.g., efficiency or on-device deployment, as discussed in [1]), this rationale should be articulated and justified.


The writing can also be improved by reducing redundancy. Integration Complexity (line 221) and Lack of Systematic Comparison (line 235) essentially describe the same problem — fragmented evaluation pipelines. These can be consolidated into one coherent discussion to improve focus and reduce redundancy.

The paper does not clearly describe how the DHARMA Classification Result is computed. Since DHARMA is presented as a “classification” metric, it is unclear why the reported results are continuous values exceeding 1. A clear mathematical formulation or example trajectory would help readers understand how these scores are derived.

The section “Implementation choices within the same architectural family create substantial security variations” is one of the most promising parts of the paper, but remains underdeveloped. It would be valuable to provide concrete examples of how specific design decisions (e.g., sub-agent autonomy, context organization, or planning strategy) increase or mitigate vulnerability. Doing so would offer actionable insights for practitioners seeking to design more secure MAS architectures. In this regard, more detailed descriptions of the implementation differences across the included frameworks (e.g., Magentic-One, LangGraph, OpenAI Agents) would make the analysis substantially more informative than the current high-level division between “centralized” and “decentralized” systems. Since the high-level ideas of them are similar, but those subtle nuances make them perform differently.

The four benchmarks covered in the experiments represent distinct threat models — such as direct injection versus indirect injection — which correspond to fundamentally different vulnerability types. The paper would benefit from an explicit comparison across these threat models, discussing which architectural designs are more robust under which attack categories. Such analysis would make the conclusions more actionable for the broader research community.

**Questions:**

See Weakness.

---

> ### Author Response · Authors · 2025-11-18
>
> We sincerely thank the reviewer for the detailed feedback on writing quality and technical clarity. We will make the necessary corrections to improve readability.
>
> To clarify, DHARMA classifies each sample and its trajectory into one of the predefined DHARMA classes (e.g., Planner-Failed, Planner-Stop, etc.). We then aggregate these classifications across all samples and report them as percentages in Tables 2 and 3. The values represent the percentage of samples falling into each class.
>
> We appreciate the recognition that Section 6.2 contains promising insights. We have provided concrete vulnerability examples in Appendix A (referenced in Section 6.2). Regarding the request for more implementation-level details: we deliberately focus on design-level differences rather than low-level implementation details (tool wrappers, message ordering, retry logic) because practitioners select entire frameworks, not individual primitives. When choosing between LangGraph and Magentic-One, users adopt complete architectural patterns—the planning approach, prompt scaffolds, and coordination mechanisms as coherent systems. Decomposing frameworks into granular implementation choices would obscure the framework-level comparisons that inform real-world deployment decisions.
>
> We thank the reviewer for this insightful suggestion and agree that an explicit comparison across threat models would strengthen our analysis and make our findings more actionable.
> Currently the datasets offer the following threat models
> - ASB: Direct Prompt Injection
> - RedCode: RedCode further involves 4 splits, Code, Code w/ jailbreak, Text Summary, Text Description. These represent diverse threat models, Direct Explicit harm, Jailbreak Prompt, Translation to NL
> - AgentHarm: Direct Explicit Harm
> - SafeArena: Direct Explicit Harm
>
> We would add a dedicated section on this to the paper.

---

### Official Review · Reviewer_Z4bc · 2025-11-01

**Soundness:** 2
**Presentation:** 2
**Contribution:** 2
**Rating:** 4
**Confidence:** 3

**Summary:**

The paper proposes SAFEAGENTS, a unified evaluation layer for multi-agent systems across popular frameworks, and DHARMA, a taxonomy for labeling where safety breaks down (planner vs sub-agent vs unmitigated execution vs error). Experiments span several benchmarks (jailbreaks, prompt-injection, code exec, web tasks) and compare single-agent vs centralized/decentralized MAS.

**Strengths:**

1. Conceptual move from “did it refuse?” to “where did it fail?” is overdue and useful in practice.

2. DHARMA’s top-level split (planner vs sub-agent) is intuitive and maps to actionable design knobs.

3. SAFEAGENTS abstraction could enable apples-to-apples comparisons across frameworks instead of the usual benchmark bingo.

4. It shows centralized planners can spread bad plans efficiently when guardrails wobble.

**Weaknesses:**

1. The DHARMA taxonomy contains internal inconsistencies that make categories non-exclusive. In particular, the leaf describing planner failure simultaneously references the absence of a valid plan and the continuation “despite a valid plan,” which is self-contradictory.

2. The paper relies on a single LLM-as-judge to assign DHARMA labels, yet it provides no human adjudication study, no second judge model, and no error bars. Because the core claims depend on these fine-grained labels, the absence of inter-rater reliability and sensitivity analyses leaves the results vulnerable to judge-specific artifacts.

3. Mechanistic claims are supported by anecdotes rather than prevalence analysis. The paper highlights illustrative traces such as sub-agent refusal being ignored or execution continuing after planning failure, but it does not quantify how frequently these patterns occur within or across datasets.

**Questions:**

1. How do you impose a deterministic, mutually exclusive labeling scheme for DHARMA on multi-turn trajectories, especially when the planner first refuses and later proceeds, and will you correct the contradictory "planner failed yet valid plan" leaf by publishing explicit state-transition rules, tie-break priorities, and worked examples that map concrete log events to exactly one label with reported human/LLM agreement?

2. What is the empirical reliability of your LLM-as-judge pipeline, and how sensitive are your conclusions to the choice of judge model? You assert manual classification is impractical and default to an LLM judge, but you never report human-human agreement, human-LLM agreement, or a second-judge ablation, even though the entire contribution hinges on these fine-grained labels.

3. Which design primitive actually causes the reported safety deltas once you remove framework confounds, and do those effects persist across different threat models? Right now you attribute differences to stratified versus combined planning and to “context organization,” but those vary alongside prompt scaffolds, retry logic, tool wrappers, and message ordering across frameworks; so far that’s correlation with a cape on.

4. Additional question on paper type: this paper reads more like a positional paper, and what concrete experimental depth elevates it beyond framework evangelism? Much of the writing emphasizes the SAFEAGENTS abstraction and the taxonomy rhetoric, with generous prose on classes, methods, and configurability, but the decisive evidence is thin where it matters most: validated labels, single-primitive causal ablations, stratified threat-model results, and uncertainty reporting. If the aim is a main-track paper, you need to move beyond “we propose a framework and a lens” into “we prove which knobs matter, how much, and when,” backed by reproducible logs and reliability metrics.

---

> ### Author Response · Authors · 2025-11-18
>
> Thank you for your valuable comments.
>
> 1. We appreciate the request for clarification on our labeling scheme. As described in Section 5.2 (line 298), we evaluate planner-level behaviors before proceeding to sub-agent-level behaviors, which ensures mutually exclusive categories through hierarchical evaluation. For multi-turn trajectories where the planner first refuses then proceeds, we classify this as Planner-Ignore. Our decision logic follows a strict precedence: we first check for refusal, if present, the trajectory is labeled either Planner-Stop or Planner-Ignore based on subsequent behavior; if no refusal occurs, we evaluate plan validity where invalid plans are labeled Planner-Fail, while valid plans proceed to sub-agent behavior classification. This hierarchical logic is encoded in our judge prompts to ensure deterministic labeling. However, we acknowledge that the current presentation lacks sufficient clarity. In the revised version, we will include: (1) explicit state-transition rules with formal precedence ordering, (2) tie-break priorities for edge cases, and (3) worked examples mapping concrete multi-turn log events to their final labels. Combined with the human-LLM agreement study mentioned in our next response point, these additions will resolve ambiguities and further demonstrate the reliability of our labeling scheme.
>
>
> 2. We appreciate this important concern about evaluation reliability. We did manually verify label accuracy for a subset of cases and employed one-shot examples tailored to each DHARMA scenario in our judge prompts, with framework-specific prompt specialization to improve classification reliability. Following on your suggestions, we will include: (1) a human-LLM agreement study with inter-rater reliability metrics comparing expert human annotations with LLM judge classifications on a representative sample, (2) a second-judge ablation using an alternative judge model to assess sensitivity of conclusions to model choice, and (3) uncertainty quantification including confidence scores and disagreement rates to contextualize our findings.
>
>
> 3.
> We appreciate this important methodological concern about confounding variables.
> Our study deliberately operates at the framework design level rather than the implementation detail level.
>
> We deliberately do not drill down to implementation details like tool wrappers, message ordering, and retry logic because doing so would defeat the purpose of comparing across different frameworks using a unified mechanism. From practioners point of view, practitioners don't assemble agents from isolated primitives—they choose frameworks. When someone selects LangGraph versus Magentic-One, they get the entire package: the planning approach, the prompt scaffolds, the tool wrappers, and the message ordering as a coherent system. If we decompose each framework into its constituent implementation choices, we lose the ability to compare different frameworks. These elements co-evolve within each framework's design philosophy and cannot be meaningfully separated in practice.
>
> - **On causality**: We don't claim to isolate a single "design primitive" causing safety deltas. Rather, we identify crucial design primitives that demonstrate measurably different security profiles across our threat model. Whether this emerges from planning decomposition itself, the message structures it necessitates, or their interaction is precisely the question our taxonomy enables.
> - **On scope**: Debugging individual implementations (e.g., "is it the tool wrapper or the planning?") is valuable but orthogonal to our contribution. Practitioners choosing between frameworks need to understand framework-level security characteristics. Researchers drilling into causal mechanisms need our taxonomy as a starting point to design controlled experiments that do isolate primitives.
>
>
> 4. We appreciate the concern about paper type and experimental rigor. This is an empirical evaluation paper. MAS are being deployed in real-world security-sensitive contexts, yet their security properties remain critically underexplored. Our contribution is systematic, comparative evaluation across popular frameworks that practitioners are already using. While we do introduce SAFEAGENTS and a taxonomy, these are means to enable the core contribution: reproducible security evaluation of frameworks.
>
> You're correct that single-primitive ablations would strengthen causal claims. However, our scope is comparative evaluation of deployed frameworks, not controlled experimentation with synthetic variants. We document that security differences exist and identify the design primitives that cause these vulnerabilities. Further causal understanding is an important subject matter of future study, but our paper provides the foundation to start teasing them apart, and provides guidance to practitioners on tools they're using today.

---

### Official Review · Reviewer_2hNM · 2025-11-02

**Soundness:** 3
**Presentation:** 3
**Contribution:** 3
**Rating:** 4
**Confidence:** 4

**Summary:**

The paper introduces SafeAgents, a unified framework for evaluating the safety of multi-agent LLM systems, and proposes DHARMA, a metric that localizes vulnerabilities (planner, sub-agent, or orchestration-level). Evaluations across five MAS architectures and four benchmarks show that common design choices can create “weak links” leading to unsafe behavior.

**Strengths:**

1. Addresses a timely and underexplored problem of security in multi-agent LLMs.
2. Broad empirical coverage across multiple architectures, datasets, and models.

**Weaknesses:**

1. The paper has limited technical contribution. The “weak links” analysis and DHARMA classes are mostly heuristic categorizations without strong theoretical grounding.
2. The paper has limited novelty. The contributions are primarily in system design and taxonomy rather than algorithmic innovation.

**Questions:**

See weakness above.

---

> ### Author Response · Authors · 2025-11-18
>
> Thank you for your valuable comments and acknowledging that this is a timely and underexplored problem.
>
> The differences in the existing agent frameworks at the level of design and their impact on security of MAS have not been explored; we undertake that study in this work.
> As rightly pointed out by you and Reviewer imRK, as the world moves from single agent to MAS, understanding the security of these systems is of paramount importance. So, the main technical contributions lie in (1) identifying the main design primitives and understanding their implications on MAS security, this is backed by (2) thorough evaluation on multiple popular benchmarks and datasets along with (3) introduction of DHARMA classification metric.
>
> We believe this provides insights for future design of secure MAS. While we do not propose new algorithms, we provide the foundational empirical understanding and evaluation methodology necessary for future algorithmic innovations in secure MAS design. Given that practitioners are already deploying these frameworks in security-sensitive contexts with no existing guidance, we believe establishing this empirical foundation represents a significant and timely contribution to the field. We are open-sourcing our code to enable future research.

---

### Official Review · Reviewer_imRK · 2025-11-07

**Soundness:** 2
**Presentation:** 3
**Contribution:** 2
**Rating:** 4
**Confidence:** 3

**Summary:**

This paper addresses the critical and under-explored security vulnerabilities of LLM-based Multi-Agent Systems (MAS). The authors posit that safety guarantees developed for single-agent (SA) systems do not transfer to MAS, as the decomposition of tasks and fragmentation of context introduce new attack vectors. Existing evaluation metrics (like ASR or ARIA) are insufficient because they only report final outcomes and cannot pinpoint why or where in the agentic pipeline a failure occurred.

Using SAFEAGENTS and DHARMA, the authors conduct a comprehensive study on five MAS architectures across four safety benchmarks. Their key findings reveal that centralized systems are not inherently safer and can be more vulnerable than single agents. They identify critical weak links tied to design choices: for instance, centralized systems that delegate only atomic instructions (low sub-agent autonomy) are less secure because this "obscures harmful objectives" from the sub-agents, leading to unmitigated execution.

**Strengths:**

The paper tackles a vital and forward-looking research gap. As the community shifts from single-agent systems to more complex multi-agent collaborations, understanding their unique security profile is paramount. The authors correctly argue that single-agent safety alignment does not guarantee MAS safety, providing a strong motivation for their work.

The paper delivers non-obvious and important insights. The central finding—that low sub-agent autonomy (atomic delegation) is a critical weak link—is a key, actionable takeaway for MAS designers.

**Weaknesses:**

1. The DHARMA metric itself is well-designed, but its implementation relies on an LLM-as-judge to classify trajectories. This introduces a significant potential source of error and non-determinism. The paper provides the (very long) prompts used for classification  but offers no validation of the judge's accuracy. A misclassification by the judge (e.g., labeling a Sub-agent-Ignored as Unmitigated Execution) could materially skew the core results in Table 2. The reliability of the paper's central metric is therefore based on an unverified assumption of the judge's correctness.

2. The analysis attributes security differences to specific "design primitives," such as sub-agent autonomy (atomic vs. high-level tasks). However, the frameworks being compared (Magentic vs. LangGraph) also differ in other primitives simultaneously, such as planning strategy (stratified vs. combined) and context organization (the literal prompt structure). The effects of these primitives are entangled. It's difficult to be certain that the vulnerability comes from "atomic delegation" (autonomy) and not, for example, simply a less safe prompt template (context organization) in Magentic. The study does not fully disentangle these variables.

**Questions:**

See weakness

---

> ### Author Response · Authors · 2025-11-18
>
> Thank you for your valuable comments.
> 1. LLM-as-Judge accuracy: The judges accuracy is indeed a very crucial dimension. To make sure that the judge is as accurate as a possible, we have included few shot examples for each of the senarios that are possible, and we have specialized the prompts for each agentic framework (separate prompts for Magentic, LangGraph and OpenAI agents). Along with this we have used a GPT-4.1 as the judge model for more accurate results. Additionally to provide concrete evidence, we will also include a study on human-LLM agreement and put experiments on a secondary LLM-as-judge to improve the robustness of the study.
>
> 2.  Multiple design dimensions:
> As you have rightly pointed out, there are multiple dimensions along which each of these frameworks vary. Interpolating these frameworks to create a synthetic family of frameworks where we can control the variation along a particular dimension at a time is possible, however, those kind of frameworks may not exist in practice. For example, a framework where the sub-agent autonomy is absent and has a combined planning phase does not exist in practice. Instead we have grounded our analysis on existing popular frameworks where multiple things are naturally changing, and then we do the analysis manually to attribute the results to the specific primitives.
>
> In order to get to the right conclusions we have followed the following protocol:
>
> - Uniform prompting for agents
>
> To maintain uniformity in prompting for agents, we have kept the system prompts to be the same for each individual sub-agents (WebSurfer, FileSurfer, etc.) and moreover for orchestrators and planning agents in each frameworks, the prompts are adapted from Magentic\-one's planning and orchestrator prompts. This avoids any discrepancies with regards to prompting.
>
> - DHARMA framework
>
> Our objective in this work is to do comparative analysis of security across multiple agent frameworks despite multitute of lower-level implementation differences. To accomplish this, we intentionally chose the classification of frameworks' weaknesses at a higher-level of abstraction than at a lower implementation level. For example, Magentic-One has stratified planning and LangGraph has combined planning, but failure in planning in either of the frameworks is classified as planning failure. This unification helps us observe patterns that generalize across frameworks.
>
> - Manual validation
>
> Given the actual classifications, we manually observe the patterns, those observations are reflected in the analysis results (Section 6.2). Whether we can  fully automate this detailed analysis in the presence of such diversity remains to be seen.

---

### Author Response · Authors · 2025-11-26
**Human-LLM Agreement Study : Response to Reviewers imRK and Z4bc**

We thank the reviewers for this critical concern. We have conducted a validation study to verify DHARMA classification reliability. We randomly sampled 100 trajectories across all four datasets. We then manually classified them using the DHARMA taxonomy, which we then compared against the LLM judge's classifications. The LLM judge achieved 95.0\% agreement with the human annotator (Cohen's Kappa = 0.899, indicating "almost perfect" agreement).

We attribute these results to our robust judge prompts. Since we designed framework-specific prompts tailored to Magentic-One, LangGraph, and OpenAI distinct logging formats (prompts are given in appendix), and included  few-shot examples per DHARMA category.

The disagreements stemmed from a specific error pattern: the LLM judge occasionally violated DHARMA's hierarchical precedence by classifying trajectories as Sub-agent Stop when they should have been Planner Ignored. This occurred in cases where both the planner and sub-agents refused, but the orchestrator ignored the planner's refusal—the LLM focused on the later sub-agent behavior rather than correctly prioritizing the planner-level refusal per DHARMA's definition.


We will expand this validation study and  include more results into the paper.

---

### Meta-Review · Area_Chair_cT8a · 2025-12-22

**Summary:**

This paper presents SafeAgents, a unified framework to assess the security vulnerabilities of multi-agent systems (MAS) that utilize LLMs. The authors introduce DHARMA, a diagnostic measure that identifies weak links in the design of MAS and evaluates performance across five architectures and four datasets. Their findings reveal that common design choices can expose MAS to significant vulnerabilities, highlighting the necessity for security-focused design in these systems, particularly in centralized configurations that delegate atomic instructions to sub-agents.

**Reviewer Concerns:**

Based on the reviewers' feedback, the paper has several identified weaknesses that could be improved upon:

- *Reliability of LLM-as-Judge*: Reviewers wished authors had validated the accuracy of the LLM used to classify trajectories, as its performance is crucial for the reliability of the DHARMA metric. **Authors agreed with this limitation; to successfully address this, they conducted a human validation on 100 trajectories, finding the LLM-judge to have 95.0% agreement with the human annotator (Cohen's Kappa = 0.899, indicating "almost perfect" agreement).**
- *Confounding Variables in Security Analysis*: Two reviewers noted that the analysis entangles multiple design primitives and did not adequately disentangle their effects, making it difficult to attribute security differences to specific factors. **Authors acknowledge this, as their focus was on testing existing MAS frameworks rather than experimentally controlling agent system prompts. However, they should tone down their claims, as confounds likely still exist due to their experimental design decisions. Further, this may not be sufficient nor convincing; they should consider adding experiments where specific features are experimentally manipulated, for more actionable insights into designing *new* frameworks.**
- *Limited Technical Contribution and Novelty*: Reviewers felt that the paper's technical contributions were limited and primarily heuristic, raising concerns about its novelty in algorithmic innovation. **Authors correctly pointed out that their investigations are more empirical than theoretical, thus making this weakness not very relevant.**
- *Internal Inconsistencies in DHARMA Taxonomy*: Reviewers highlighted that the DHARMA taxonomy contains conflicting categories that undermine its clarity and utility. **Authors acknowledged the lack of clarity and will specify that their hierarchical labeling indeed creates mutually exclusive categories.**
- *Lack of Mechanistic Prevalence Analysis*: Reviewers pointed out that claims about mechanisms were anecdotal rather than quantitatively analyzed, which diminishes their robustness. **Authors did not address this weakness directly.**
- *Writing and Clarity Issues*: Reviewers felt that the paper requires significant improvements in writing to enhance its clarity and understanding of concepts such as the DHARMA Classification Result. **Authors will provide more details on this in the paper.**
- *Comparison Across Distinct Threat Models*: The reviewers felt that an explicit comparison across different threat models would strengthen the paper's conclusions and make them more actionable for the broader research community. **Authors agree that such an analysis would strengthen their work and will add a section in the paper.**

**Reviewer Scores:**

They might have improved!

---

### Decision · Program_Chairs · 2026-01-26

Reject